# Development of a Machine Learning Model to Distinguish between Ulcerative Colitis and Crohn’s Disease Using RNA Sequencing Data

**DOI:** 10.3390/diagnostics11122365

**Published:** 2021-12-15

**Authors:** Soo-Kyung Park, Sangsoo Kim, Gi-Young Lee, Sung-Yoon Kim, Wan Kim, Chil-Woo Lee, Jong-Lyul Park, Chang-Hwan Choi, Sang-Bum Kang, Tae-Oh Kim, Ki-Bae Bang, Jaeyoung Chun, Jae-Myung Cha, Jong-Pil Im, Kwang-Sung Ahn, Seon-Young Kim, Dong-Il Park

**Affiliations:** 1Division of Gastroenterology, Department of Internal Medicine and Inflammatory Bowel Disease Center, Kangbuk Samsung Hospital, Sungkyunkwan University School of Medicine, Seoul 03181, Korea; skparkmd@gmail.com; 2Medical Research Institute, Kangbuk Samsung Hospital, Sungkyunkwan University School of Medicine, Seoul 03181, Korea; chilwoo.lee@gmail.com; 3Department of Bioinformatics, Soongsil University, Seoul 06978, Korea; sskimb@ssu.ac.kr (S.K.); gy48085@gmail.com (G.-Y.L.); ssky1324@naver.com (S.-Y.K.); rladhkschlrh@gmail.com (W.K.); 4Personalized Medicine Research Center, Korea Research Institute of Bioscience and Biotechnology (KRIBB), Daejeon 34141, Korea; nlcguard@kribb.re.kr; 5Department of Internal Medicine, College of Medicine, Chung-Ang University, Seoul 04388, Korea; gicch@cau.ac.kr; 6Department of Internal Medicine, College of Medicine, Daejeon St. Mary’s Hospital, The Catholic University of Korea, Daejeon 34943, Korea; dxandtx@catholic.ac.kr; 7Department of Internal Medicine, Haeundae Paik Hospital, Inje University College of Medicine, Busan 48108, Korea; kto0440@paik.ac.kr; 8Department of Internal Medicine, Dankook University College of Medicine, Cheonan 31116, Korea; kibaebang@gmail.com; 9Department of Internal Medicine, Gangnam Severance Hospital, Yonsei University College of Medicine, Seoul 06273, Korea; J40479@gmail.com; 10Department of Internal Medicine, Kyung Hee University Hospital at Gang Dong, Kyung Hee University College of Medicine, Seoul 05278, Korea; drcha@khu.ac.kr; 11Department of Internal Medicine and Liver Research Institute, College of Medicine, Seoul National University, Seoul 03080, Korea; jpim0911@snu.ac.kr; 12Functional Genome Institute, PDXen Biosystems Inc., Daejeon 34129, Korea; kwangsung.ahn@gmail.com

**Keywords:** inflammatory bowel disease, Crohn’s disease, ulcerative colitis, RNA sequencing, machine learning

## Abstract

Crohn’s disease (CD) and ulcerative colitis (UC) can be difficult to differentiate. As differential diagnosis is important in establishing a long-term treatment plan for patients, we aimed to develop a machine learning model for the differential diagnosis of the two diseases using RNA sequencing (RNA-seq) data from endoscopic biopsy tissue from patients with inflammatory bowel disease (*n* = 127; CD, 94; UC, 33). Biopsy samples were taken from inflammatory lesions or normal tissues. The RNA-seq dataset was processed via mapping to the human reference genome (GRCh38) and quantifying the corresponding gene models that comprised 19,596 protein-coding genes. An unsupervised learning model showed distinct clusters of four classes: CD inflammatory, CD normal, UC inflammatory, and UC normal. A supervised learning model based on partial least squares discriminant analysis was able to distinguish inflammatory CD from inflammatory UC after pruning the strong classifiers of normal CD vs. normal UC. The error rate was minimal and affected only two components: 20 and 50 genes for the first and second components, respectively. The corresponding overall error rate was 0.147. RNA-seq analysis of tissue and the two components revealed in this study may be helpful for distinguishing CD from UC.

## 1. Introduction

Inflammatory bowel disease (IBD) is a chronic intestinal disease that is multifactorial and polygenic, resulting from the dysregulation of the mucosal immune response and intestinal microflora [1]. Recent genome-wide association studies of IBD, including both Crohn’s disease (CD) and ulcerative colitis (UC), have led to the identification of a number of susceptibility genes or loci associated with CD and UC [2,3]. Over 200 distinct susceptibility loci for IBD were identified, with some contributing to a unique risk and others contributing to a combined risk for CD and UC [2,4]. In fact, approximately 70% of the genes are shared between CD and UC, thereby highlighting a significant genetic overlap in these disease entities [2]. These results strengthen earlier suggestions that CD and UC are, at the molecular level, two related yet different forms of chronic intestinal disease. However, UC and CD are heterogeneous, with marked differences in disease location, behavior, clinical presentation, and response to treatment [5,6,7].

Emerging challenges have now moved from gene identification to functional understanding. RNA sequencing (RNA-seq) is a technique that can examine the quantity and sequences of RNA in a sample using next-generation sequencing (NGS). It analyzes the transcriptome, which is key for connecting the information in our genome with its functional protein expression [8]. With high-throughput omics data such as proteomics, epigenomics, metabolomics, and transcriptomics, powerful tools such as artificial intelligence, including machine learning and deep learning, are emerging to unravel novel mechanistic insights and help address unmet clinical needs in IBD, including diagnosis, disease risk assessment, and prognosis [9].

As previous GWAS emphasized a substantial overlap in the genetic risk loci between UC and CD [2], and previous RNA-seq transcription profiles in the specific colon segment also reflected a “common IBD” signature with a major overlap between the UC and CD [10], the need for a new method to discriminate between UC and CD using high-throughput omics data emerged.

In this study, we profiled gene expression in inflammatory or normal mucosal samples obtained from patients with IBD using RNA-seq with the aim of differentiating between UC and CD using a machine learning model.

## 2. Materials and Methods

### 2.1. Study Population

In our study, we utilized two cohorts of patients. Patients with CD were included from the IMPACT (identification of the mechanism of the occurrence and progression of CD through integrated analysis on both genetic and environmental factors) study cohort, which is a prospective multicenter study established in Korea in 2017 [11,12]. A total of 16 university hospitals participated in this study, and clinical data and biological specimens (including blood, stool, and tissue specimens) of CD patients, who were newly diagnosed or followed-up within the institutions, were collected. Patients with UC were included from the UC multiomics study cohort, which is a prospective multicenter study established in Korea in 2020. A total of 14 university hospitals participated in this study, and collected clinical data and biological specimens (including blood, stool, tissue, and saliva specimens) of UC patients. Ethical approval of the present study was provided by the institutional review boards of Kangbuk Samsung Hospital (KBSMC 2016-07-029, KBSMC 2020-05-021) and each center. Written consent was obtained from all participants after the nature and possible consequences of the studies were explained.

### 2.2. Sample Collection

RNA-seq data were collected from the colonoscopic tissue samples of patients with CD and UC. Patients previously diagnosed with UC or CD underwent colonoscopy as part of their routine medical care. Biopsy was performed from the most severe inflammatory lesion or normal lesion. Colon tissue was assessed as inflamed or normal by the performing endoscopist based on endoscopic findings at the time of collection. Normal tissue was obtained mostly from the terminal ileum of CD patients and the rectum of UC patients when the patient’s colonoscopic findings showed endoscopic remission.

### 2.3. RNA Extraction, Library Construction, and Sequencing

Colon tissue for RNA-seq analysis was transferred to individual tubes containing RNA, later (Invitrogen, Waltham, MA, USA, AM7021), stored at 4 °C overnight, and then stored at −80 °C until processing. The samples were added to TRI-Reagent (Molecular Research Center, Inc., Cincinnati, OH, USA, TR118) for lysis of the tissue. The samples were homogenized, and chloroform was added to the samples containing TRI-Reagent. The samples were vortexed and centrifuged, and 70% ethanol was added to the supernatant. Total RNA was treated with the RNase-free DNase Kit (Invitrogen, AM1906) to remove contaminating DNA from the RNA preparations. Total RNA was re-suspended in 30 μL of RNA/DNA-free water.

The quality and quantity of the extracted total RNA were analyzed using an ND-1000 spectrophotometer (Thermo Fisher Scientific, Waltham, MA, USA), and a 2100 Agilent Bioanalyzer (Agilent Technologies, Waldbronn, Germany) was used to estimate the RNA integrity number (RIN) score. Samples were only used in subsequent analyses if the RNA integrity number (RIN) was greater than 5.5.

Approximately 1 μg of total RNA was used for library construction with the Illumina TruSeq Stranded Total RNA Library Prep Kit (San Diego, CA, USA). Next, paired-end sequencing was performed using the Illumina NovaSeq 6000 System, according to the manufacturer’s instructions, yielding 101-bp paired-end reads. All primary RNA sequencing data were deposited in the SRA database under the accession number PRJNA774132.

### 2.4. RNA Sequencing Data Analysis

The sequenced reads were mapped to the human reference genome GRCh38 using HISAT2 [13]. The gene-level expression was quantified with featureCounts [14] from the Subread package using the Ensembl gene model (Release 100). The read counts were processed using the Bioconductor R package edgeR [15]. Specifically, the filtration of genes with very low counts across all samples and the calculation of sample-specific effective library sizes for normalization were performed with the default options of edgeR. For the analysis of differentially expressed genes, the negative binomial generalized linear model was fitted to each gene with naïve Bayesian dispersion and a design matrix of two factors (disease and inflammatory status), followed by a quasi-likelihood F-test comparing all four combinations: inflammatory vs. normal of each disease, as well as CD vs. UC of inflammatory or normal tissues. In addition, edgeR was used to calculate the log2-transformed normalized counts, which were input into the principal component analysis (PCA), heatmap drawing, and machine learning algorithms. The Bioconductor R package mixOmics [16] was used to draw the heatmaps.

### 2.5. Machine Learning for Prediction Model Development

The Bioconductor R package mixOmics was used for PCA and prediction model development. We applied partial least-squares discriminant analysis (PLS-DA), which is computationally efficient with rich graphical outputs, in two stages: first classifying CD normal vs. UC normal, followed by pruning strong classifiers and classifying inflammatory CD vs. inflammatory UC. The strong classifiers of normal samples were identified by calculating the loading vector of each gene along the direction connecting the class means. From the sorted list of the vector lengths, we tested three trenches that were cut into 1–3 quartiles. We confirmed that the lower quartiles, that is, those with more pruning, showed poorer performance in classifying normal CD versus normal UC.

While pruning more genes in this way worsens the classification performance of inflammatory CD vs. inflammatory UC slightly, we pruned half of the genes and used the rest in the subsequent classification of inflammatory CD vs. inflammatory UC.

The classification performance of the PLS-DA model was assessed by a five-fold cross-validation (5-CV), which was repeated 10 times with random splits. The optimum number of components was determined from the trend of the overall error rate of 5-CV. A sparse PLS-DA model was developed for the classification of inflammatory CD vs. inflammatory UC. sPLS-DA enables the selection of the most predictive or discriminative features in the data to help classify the samples. sPLS-DA tuned the number of genes from 1 to 300 for each of the components suggested by the previous PLS-DA. The performance of this process was also evaluated using 5-CV, which was repeated 100 times with random splits. The final model was then defined using the optimum number of genes for each of the optimum number of components.

## 3. Results

The sample collection consisted of 94 CD and 33 UC patients. The demographic and clinical details of the patients are provided in Appendix A. The number of inflammatory and normal samples was 29 and 65 in CD and 20 and 13 in UC, respectively.

### 3.1. Unsupervised Learning via PCA

To investigate mucosal pathophysiology in IBD, we first performed an unsupervised PCA analysis of the RNA-seq data (Figure 1). PCA indicates sample similarity/differences based on all the data included. The PCA of the log-transformed expression values revealed the distinct clustering of samples by the presence of inflammation and disease type. The first component (39%) mostly separated the inflammatory samples from the normal samples, and the second component (9%) roughly distinguished CD from UC.

### 3.2. Supervised Learning of Differentially Expressed Genes (DEGs) and Pathway Analyses

Given the distinct clustering patterns observed in an unsupervised PCA analysis, we next used the supervised learning of DEG to assess the relative contribution of disease type and inflammation in mucosal gene expression profiles. With a fold change of greater than 2 and false discovery rate (FDR) < 0.05, 1966 genes were upregulated and 979 genes were downregulated in inflamed CD mucosa compared with normal CD mucosa. In inflamed UC mucosa, 1498 genes were upregulated and 571 genes were downregulated compared with normal UC mucosa. When comparing inflamed CD with inflamed UC, there were 1050 and 395 genes that showed higher and lower expressions, respectively. There were 638 and 425 genes that were upregulated and downregulated, respectively, between the normal CD and UC groups. A full list of DEGs is available in Appendix A.

To gain an insight into the biological roles of the most significant up- or downregulations, KEGG pathway analysis was performed using the DAVID web service [17] (Figure 2). The common pathways of upregulated genes in CD-inflammatory and UC-inflammatory samples were compared with CD- and UC-normal samples; these included: (1) Adaptive immunity: IBD; (2) Signal transduction: NF-kappa B signaling, TNF signaling, JAK-STAT signaling, and cytokine-cytokine receptor interaction; (3) Adhesion and differential: hematopoietic cell lineage and osteoclast differentiation; and (4) Autoimmune disease-related rheumatoid arthritis. There were 22 pathways with upregulated genes in CD-inflammatory compared with UC-inflammatory samples, suggesting CD-specific pathogenesis. See Appendix A for further details.

### 3.3. Classification Models Based on PLS-DA and sPLS-DA

Since unsupervised learning showed distinct clusters of four classes—CD inflammatory, CD normal, UC inflammatory, and UC normal—and the supervised learning of DEG and their common pathways in inflammatory samples presented their association with IBD pathogenesis, we sought to develop a supervised classification model. We aimed to distinguish inflammatory CD from inflammatory UC after pruning the strong classifiers of normal CD vs. normal UC based on PLS-DA. First, PLS-DA was applied to our dataset of 65 normal CD and 13 normal UC samples, and the corresponding PLS loading along the direction connecting the class centers was calculated for each gene. Based on the loading values, the top 1/4, 1/2, and 3/4 of the genes were pruned, and the corresponding PLS-DA error rates were evaluated as 0.038, 0.118, and 0.174, respectively. When these trenches were applied to the PLS-DA runs of classifying 29 inflammatory CD vs. 20 inflammatory UC samples, the error rates decreased slightly to 0.078, 0.084, and 0.098, respectively. Based on these results, we pruned the top half of the genes. With the other half of the genes used in the PLS-DA of inflammatory CD vs. inflammatory UC, the corresponding overall error rate was minimal for the four components.

Because this result applies to half of all the genes, the sparse version of PLS-DA was employed to remove redundant genes and simplify the model. For each of the four components, the error rate was minimal with only two components: 20 and 50 genes for the first and second components, respectively; the gene lists are presented in Appendix A. For the two components, CD-inflammatory and UC-inflammatory genes were well-differentiated, as shown in the PLS projection (Figure 3a) and the cluster heatmap (Figure 3b). The performance of the final model was evaluated by averaging 100 5-CV runs. The corresponding overall error rate was 0.147, whereas the average error rate for each class was 0.155.

## 4. Discussion

In this study, initial unsupervised analysis of mucosal RNA-seq profiles revealed highly inflammation-specific signatures and suggested disease-associated alterations in gene expression. Given the supervised learning of DEGs, common pathways in inflammatory samples are associated with IBD pathogenesis. In addition, pathways of upregulated genes in CD inflammation, compared with UC inflammation, suggested CD-specific pathogenesis. In the supervised classification model for distinguishing CD from UC, sPLS-DA revealed two components (20 and 50 genes), and inflammatory CD and inflammatory UC were well-differentiated within these two components.

Numerous previous studies led to the identification of a number of susceptibility genes or loci associated with CD and UC. Recent GWASs identified a total of 163 IBD susceptibility loci, which focus the attention on positional candidate genes involved in immunoregulation and microbial homeostasis [2]. However, GWAS emphasized a substantial overlap in the genetic risk loci between UC and CD. To further characterize the molecular pathways underlying disease development, omics approaches such as proteomics, epigenomics, metabolomics, and transcriptomics emerged to unravel novel mechanistic insights and help address unmet clinical needs in the IBD field. Among the omics data, we used RNA-seq data and transciptomics to examine molecular pathways and to differentiate between UC and CD.

A few studies analyzed RNA-seq data in the mucosal tissues of patients with IBD. Noble et al. [18] performed a genome-wide expression study with 67 patients with UC and 31 healthy controls (HC). When expression signals were compared between 35 inflamed and 22 non-inflamed UC biopsies, 700 genes showed a fold change greater than 1.5. In our study, 1498 genes were upregulated (>2-fold) in inflammatory UC compared with normal UC mucosa, and while some genes overlapped with Noble’s results, the list of notably upregulated genes was different. Holgersen et al. [19] quantified the expression of 115 selected genes that were known to be linked to IBD. They investigated the inflamed mucosal tissues of 9 CD, 10 UC, and 15 HC. Although they compared inflammatory CD or UC tissue with HC, notably upregulated genes in CD, such as *IL1B*, *CXCL1*, and *CXCL2*, were also present in our list of the genes upregulated in inflammatory CD compared with normal CD.

Howell et al. [10] elaborated on these findings. They obtained mucosal biopsies collected from children (median age, 13 years) newly diagnosed with IBD (43 with CD, 23 with UC) and 30 HC. They investigated gut-segment-specific differences and disease-specific alterations considering inflammation status in the transcription profiles of intestinal epithelial cells (IECs). In addition to a number of genes that were found to be differentially expressed between the IBD and control samples (e.g., *DEFA5*, *DEFA6*, *LYZ*, *PLA2G2A*, *CD40*, and *CD44*), ileal IECs revealed CD-specific changes in gene expression when compared with either controls or UC. However, the changes observed in the sigmoid colon reflected a “common IBD” signature, with a major overlap between the UC and CD signatures. To overcome this issue, we applied PLS-DA, a machine learning analysis to distinguish CD from UC.

Using PLS-DA we found that two components, 20 and 50 genes, can effectively discriminate CD from UC, regardless of the gut segment. Interestingly, most genes included in the first and second components did not overlap with the DEGs of inflammatory CD compared with inflammatory UC. For example, among the first 20 components, only one gene, *complement C1q B chain* (*C1QB*), was included in the DEG list of inflammatory CD vs. inflammatory UC, and the remaining 19 genes were not included. This suggests that a better discernment of the combination of genes and components specific to CD was achieved by PLS-DA.

The present result, with a relatively large IBD cohort, allowed us to gain a valuable insight into the differences in gene expression in mucosal tissue along the disease phenotype and inflammation status. In addition, as it is frequently challenging to differentiate CD from UC for those with less severe disease in real-world practice, the two components revealed by our machine learning analysis could be of clinical value. The ability to use endoscopic mucosal biopsies rather than resection samples will allow investigators to assess a larger range of patients, encompassing those with mild to moderate disease.

However, this study has several limitations. First, the predictive ability of this model was not validated in an independent cohort. Thus, further research with a multicenter IBD cohort is ongoing. Second, although UC was more likely to reveal the diseases processes resulting in the inflammation of the regional mucosa, and previous studies showed that cluster analysis presented differences in gene expression between gut segments [18], we could not assess the relative contribution of diagnosis and inflammation to the observed variance within each data layer via the gut segments. However, before classifying inflammatory CD vs. inflammatory UC, we first classified CD normal vs. UC normal, as most of the normal samples in CD and UC were collected from the terminal ileum and rectum, respectively. Thus, in the final sPLS-DA model, the effect of the gut segment may have been minimal. Third, most genes included in the first and second components did not overlap with the DEGs of inflammatory CD compared with inflammatory UC, which was associated with the pathogenesis of CD. Thus, the mechanism of how the combination of genes can discriminate CD and UC and the role of each gene in the component needs to be further studied.

## 5. Conclusions

In conclusion, using machine learning techniques for the RNA-seq analysis of endoscopic mucosal tissues, we were able to successfully classify CD from UC. Further research on omics data with advanced sequencing technologies and bioinformatics analysis could improve our understanding of IBD pathogenesis and outcomes.

## Figures and Tables

**Figure 1 diagnostics-11-02365-f001:**
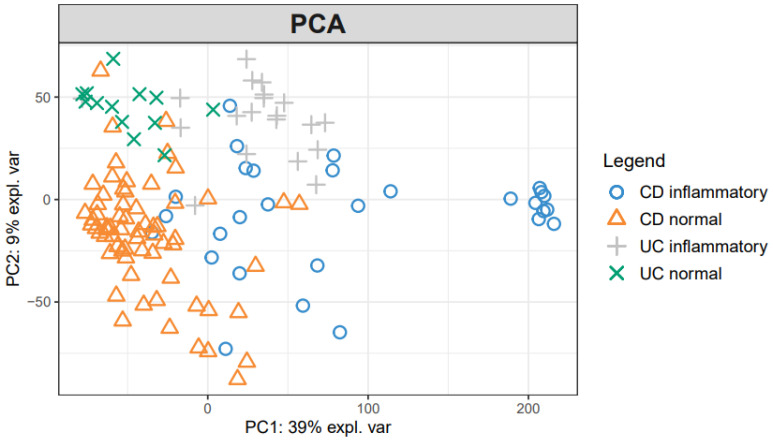
Principal component analysis plot of IBD samples. Normalized log-transformed expression values from edgeR were used in the PCA up to 10 components. The first two components that explained 48% of the total variance are shown in the plot drawn with the Bioconductor R package *mixOmics*.

**Figure 2 diagnostics-11-02365-f002:**
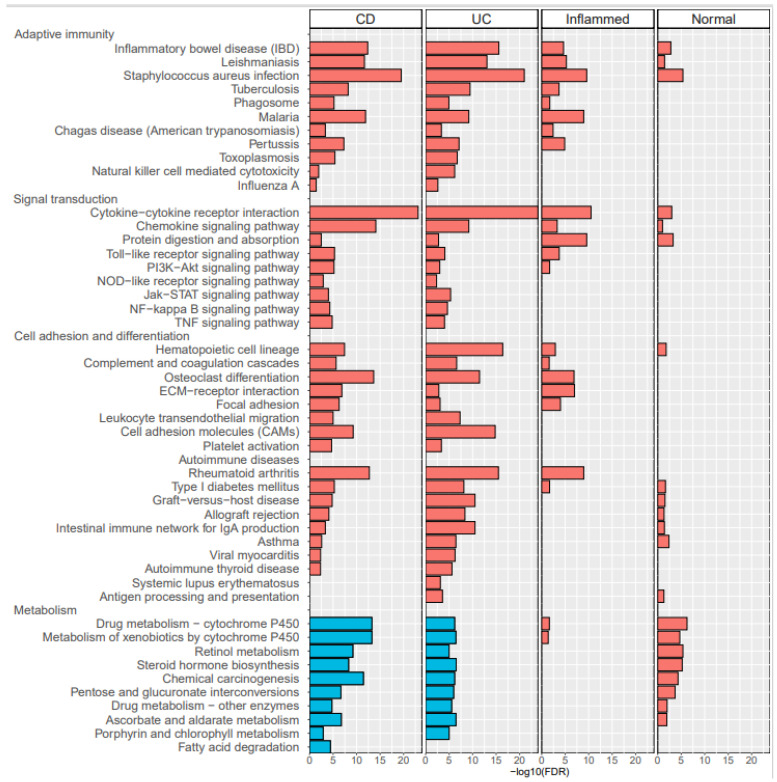
KEGG pathways enriched in the lists of differentially expressed genes (DEGs). The four panels correspond to the DEGs of inflammatory CD vs. normal CD (labeled “CD”), inflammatory UC vs. normal UC (labeled “UC”), inflammatory CD vs. inflammatory UC (labeled “Inflamed”), and normal CD vs. normal UC (labeled “Normal”). The horizontal bars represent the -log10(FDR) values from the pathway enrichment analysis calculated with DAVID web service (red for upregulated DEGs, and blue for downregulated DEGs). Note that the lists are shown for FDR < 0.05. There were no significantly enriched pathways for the downregulated genes of both “Inflammatory” and “Normal” panels. See Appendix A for details of DEGs categorized to each pathway.

**Figure 3 diagnostics-11-02365-f003:**
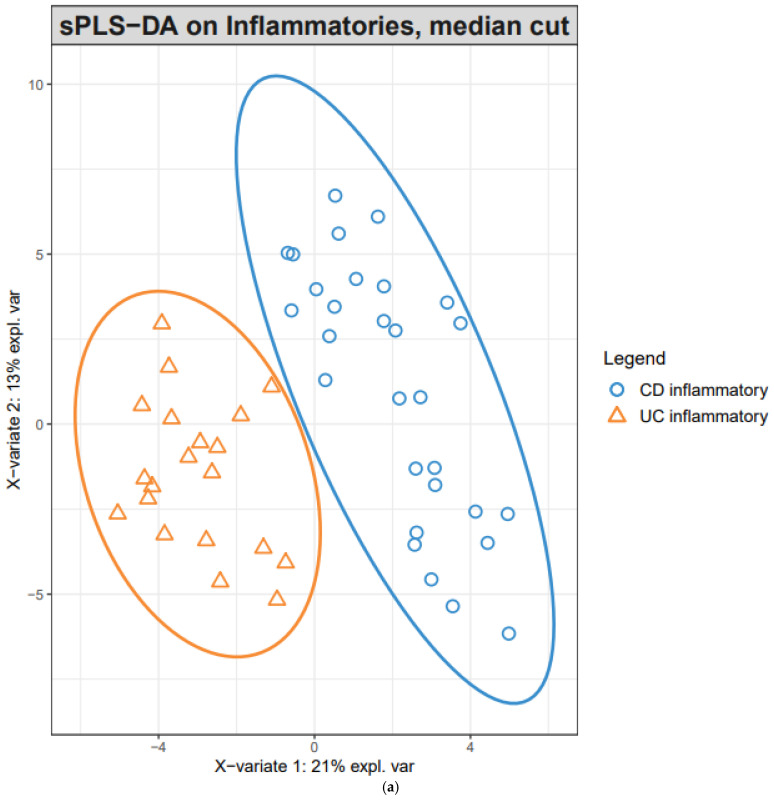
The clustering diagrams of the final sparse partial least-squares discriminant analysis (sPLS-DA) model that classifies inflammatory CD vs. inflammatory UC. (**a**) The PLS projection onto the subspace spanned by the two components. The ellipses for each class represent 95% confidence level of discrimination. (**b**) The heatmap hierarchical clustering of 70 genes (rows) and 49 samples (columns). The bottom 20 and top 30 genes belong to component 2, while the middle 20 genes belong to component 1. All plots were drawn with the Bioconductor R package *mixOmics*.

## Data Availability

All primary RNA sequencing data were deposited in the SRA database under the accession number PRJNA774132 (https://www.ncbi.nlm.nih.gov/bioproject/PRJNA774132, accessed on 10 December 2021).

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
