# Peer review of "Development of a Machine Learning Model to Distinguish between Ulcerative Colitis and Crohn’s Disease Using RNA Sequencing Data"

_diagnostics, 2021, doi:10.3390/diagnostics11122365_

Round 1

Reviewer 1 Report

Your manuscript provides interesting results on a novel approach to genetically distinguish CD from UC by using -omics and machine learning methods. I have no specific recommendation for revisions.

Author Response

Your manuscript provides interesting results on a novel approach to genetically distinguish CD from UC by using -omics and machine learning methods. I have no specific recommendation for revisions.

Respond: Thank you for your constructive review.

Reviewer 2 Report

The manuscript attempts to use RNA-sequencing via machine learning model to differentiate between UC and CD in terms of immunoregulation and microbial homostasis. The scholarship exhibited is excellent as is the presentation of the data. The design flaw is built into the data. UC, is more likely than not  collection of diseases processes resulting in inflammation of the regional mucosa. By use of rectal biopsy only for the identification of ongoing disease, the authors largely circumvent this concern. What was not programed into the study is the mechanism for disease induction. In CD, is the Th1 proinflammatory is the response to Mycobacterium avium subspecies paratuberculosis  (a mycobacterium) at its sites of attachment and antigen processing. In the various forms of UC,  the molecular pathways follow the dictates of the inciting mechanism, be it bacterial dysbiosis or enterotoxic induction.

What concerns me is the deletion data to validate a point. It blemishes a fine piece of investigation that uses a novel approach.

One problem with artificial intelligence is that it forgets to ask why or appleal to common sense, e.g. all disease are a partial function of genetic makeup of the individual.

Author Response

** We attached wrong file but the file could not be deleted. Thus please see the reply in this box.

The manuscript attempts to use RNA-sequencing via machine learning model to differentiate between UC and CD in terms of immunoregulation and microbial homostasis. The scholarship exhibited is excellent as is the presentation of the data.

  1. The design flaw is built into the data. UC, is more likely than not collection of diseases processes resulting in inflammation of the regional mucosa. By use of rectal biopsy only for the identification of ongoing disease, the authors largely circumvent this concern.

Response: Thank you for your valuable comments and we agree with your comments. UC is more likely than not collection of diseases processes resulting in inflammation of the regional mucosa and previous studies have shown that cluster analysis presents differences in gene expression between gut segments in UC. Although we classify CD normal vs. UC normal first in the PLS-DA model to overcome this issue, we could not assess the relative contribution of diagnosis and inflammation to the observed variance within each data layer by gut segments. Thus we revised the sentence in the DISCUSSION section of revised manuscript as follows; “Second, although UC is more likely than not collection of diseases processes resulting in inflammation of the regional mucosa, and previous studies have shown that cluster analysis presents differences in gene expression between gut segments, we could not assess the relative contribution of diagnosis and inflammation to the observed variance within each data layer by gut segments. However, before classifying inflammatory CD vs. inflammatory UC, we first classified CD normal vs. UC normal, as most of the normal samples in CD and UC were collected from the terminal ileum and rectum, respectively. Thus, in the final sPLS-DA model, the effect of the gut segment may have been minimal.”

  1. What was not programed into the study is the mechanism for disease induction. In CD, is the Th1 proinflammatory is the response to Mycobacterium viumsubspecies paratuberculosis  (a mycobacterium) at its sites of attachment and antigen processingIn the various forms of UC, the molecular pathways follow the dictates of the inciting mechanism, be it bacterial dysbiosis or enterotoxic induction.

Response: Thank you for your valuable comments. In this study, we focused on the new method, machine learning analysis to discriminate between UC and CD. Using PLS-DA we found that two components, 20 and 50 genes, can discriminate CD from UC. Interestingly, most genes included in the first and second components did not overlap with the DEGs of inflammatory CD compared with inflammatory UC, which was associated with mechanism of disease induction (supplementary table 3). Although it suggests that a better discernment of the combination of genes and components specific to CD was achieved by PLS-DA, the mechanism how the combination of genes can discriminate CD and UC and the role of each gene in the component were not known yet. Thus further functional study is needed. We added these contents in the DISCUSSION section of revised manuscript as follows: “Third, most genes included in the first and second components did not overlap with the DEGs of inflammatory CD compared with inflammatory UC, which was associated with pathogenesis of CD. Thus, the mechanism how the combination of genes can discriminate CD and UC and the role of each gene in the component needs to be further studied.”

  1. What concerns me is the deletion data to validate a point. It blemishes a fine piece of investigation that uses a novel approach.

Response: Thank you for your valuable comments. One may be legitimately concerned that removing too many genes might deteriorate the prediction performance.  When we removed the genes showing differential expression between normal CD vs normal UC, we made sure that the power to distinguish inflammatory CD vs inflammatory UC was not compromised. As far as the prediction is concerned, there are a multitude of genes that can perform well when they are appropriately combined. The prime objective is for developing a sparse model that can achieve good performance with a minimum number of genes. It is likely that the model we propose here may not be the only model. In other words, there may be more models with similar performance. 

  1. One problem with artificial intelligence is that it forgets to ask why or appleal to common sense, e.g. all disease are a partial function of genetic makeup of the individual.

Response: We agree that the result of an artificial intelligence application is kind of a black box, difficult to understand why those features are selected. If we had applied machine learning without checking the biological relevance of our primary data, RNA-seq, it would be difficult to appeal to common sense. We were keen on this issue and did differential gene expression analysis followed by gene set enrichment analysis to ramify the biological underpinning thereof. It was only after this confirmation that our RNA-seq dataset concur with known biology of inflammation, that we went ahead to select a few biomarker genes whose combination can distinguish inflammatory CD vs inflammatory UC.

This manuscript is very interesting, but the Academic Editor suggest to revise the following comment::
(1) In the introduction, the contribution of this paper and (2) the difference from other works should be clearly presented.

Response: Thank you for your valuable comments. As your comment, we added the sentence regarding the contribution of this paper and the difference from other works in the INTRODUCTION section of revised manuscript as follows: “As previous GWAS emphasized a substantial overlap in the genetic risk loci between UC and CD [1], and previous RNA-seq transcription profiles in the specific colon segment also reflected a “common IBD” signature, with a major overlap between the UC and CD [2], the need for a new method to discriminate between UC and CD using high-throughput omics data has emerged.

(3) Reference citations in this paper are not appropriate for the Applied Sciences form and must be rewritten. For example, the quote on page 2, "..., and response to treatment.[5] [6] [7]" is ".., and response to treatment [5] [6] [7]".

Response: Thank you for your comments. We revised reference citations in appropriate form in the revised manuscript.